# Research of the impact of economic decline on air quality in Wuhan under COVID-19 epidemic

Junda Qiu, Peng Li*, Congzhe You, Honghui Fan

School of Computer Engineering, Jiangsu University of Technology, Changzhou, China

* 372223283@qq.com

**Data Availability Statement:** The data link: http://aqicn.org/city/wuhan/cn/.

**Funding:** Funding Information: Funder Name Grant Number Grant Recipient National Natural Science Foundation of China 61806088 Honghui Fan

## Abstract

A novel economic impact model is proposed by this paper to analyze the impact of economic downturn on the air quality in Wuhan during the epidemic period, and to explore the effective solutions to improve the urban air pollution. The Space Optimal Aggregation Model (SOAM) is used to evaluate the air quality of Wuhan from January to April in 2019 and 2020. The analysis results show that the air quality of Wuhan from January to April 2020 is better than that of the same period in 2019, and it shows a gradually better trend. This shows that although the measures of household isolation, shutdown and production stoppage adopted during the epidemic period in Wuhan caused economic downturn, it objectively improved the air quality of the city. In addition, the impact of economic factors on $PM_{2.5}$, $SO_2$ and $NO_2$ is 19%, 12% and 49% respectively calculated by the SOMA. This shows that industrial adjustment and technology upgrading for enterprises that emit a large amount of $NO_2$ can greatly improve the air pollution situation in Wuhan. The SOMA can be extended to any city to analyze the impact of the economy on the composition of air pollutants, and it has extremely important application value at the level of industrial adjustment and transformation policy formulation.

## Introduction

In the late industrial and developed stage, with the increase of population density and the rapid development of economy, air pollution in many countries, including China, has become more and more serious, which has attracted extensive attention from the scientific community [1, 2]. In 2008, more than 200 scientists from 35 countries on four continents participated in the International Symposium on air quality management. At the meeting, a wide range of discussions were conducted on air quality assessment, practice, and selection of evaluation standards in urban and regional areas, which had far-reaching impact [3].

The relationship between economic growth and air quality has been one of the hot issues in the field of environmental economy. In recent years, more and more research results are reflected in the profound revelation of the quantitative relationship between various economic and air pollutant factors. Filonchyk studied the degree to which lockdown affected air quality

National Natural Science Foundation of China 62002142 Peng Li National Natural Science Foundation of China 61902160 Congzhe You Science and Technology Plan Project of Changzho CJ20220055 Junda Qiu Natural Science Foundation of Jiangsu Province BK20201057 Peng Li The funders had role in study design, data collection and analysis, decision to publish, or preparation of the manuscript.

**Competing interests:** The authors have declared that no competing interests exist.

in Shanghai and the surrounding Yangtze River Delta in China, and pointed out that human economic activities will have a direct impact on air quality [4]. Sulaymon studied the impact of lockdown on air quality as a result of the COVID-19 pandemic in Wuhan was evaluated by comparing the concentrations of the six criteria air pollutants during January 1 to June 20 from 2017 to 2020, and put forward environmental governance suggestions [5]. Aristotelous studied the relationship between economic growth and air pollution based on unbalanced panel data sets from 1961 to 2010 for different groups of countries. The results show that the relationship between economic growth and air pollution depends on the level of economic development of a country [6]. Ouyang et al. Studied the nonlinear effects of environmental regulation and economic growth on $PM_{2.5}$ in 30 OECD countries by using the panel threshold model, and discussed the main socio-economic factors driving $PM_{2.5}$ emissions. The analysis results show that with the increase of environmental policies, $PM_{2.5}$ emissions first rise and then have no significant correlation. If the current trend continues, emissions are expected to decrease [7]. Soukiazis et al. Investigated the relationship between renewable energy, economic development and pollution from 2004 to 2015 based on the strong evidence of a group of 28 OECD countries, and used the simultaneous equation system method to describe the relationship between economic variables and renewable energy and pollution emissions and feedback effect [8]. Urhie et al. explained the interaction among economic growth, air pollution and health performance by using the moderate mediation model and the analysis tool process macro developed by Hayes [9]. Sarkodie explained the trend of causality, and verifies the feedback hypothesis between material consumption and economic growth by using the new Breitung-Candelon Spectral Granger-causality relationship. Research shows that the consumption of metal ore can predict economic growth, income level and renewable energy consumption, but it will cause environmental air pollution [10].

In recent years, due to the rapid development of China's economy, a lot of air pollution problems have been brought about, which has attracted great attention of the academic community. Many researchers take China's economic growth and air pollution as the research background, trying to find the potential law between economic growth and air pollution [11, 12]. Yang et al. Studied the relationship between China's economic growth, environmental pollution, and energy consumption. The results show that there is a long-term stable cointegration relationship between China's economic growth and environmental pollution and energy consumption, and there is a long-term two-way causal relationship between economic growth and energy consumption. Energy consumption is a strong one-way causal relationship of environmental pollution [13]. Based on China's high-resolution population density map, $PM_{2.5}$ concentration retrieved by satellite and provincial health data, Xie et al. Investigated the relationship between economic development and haze and haze events in the first 10 years. In some inland developing provinces with high population density (such as Henan, Anhui and Sichuan), haze pollution in these areas was caused by more pollution and resource intensive industries the staining showed an increasing trend [14]. Lu et al. analyzed the panel data of 30 provinces in China from 2002 to 2014 and used simultaneous equation model (SEM) to describe the relationship among economic development, environmental quality public health by using, and verified the negative impact of economic development on the environment [15]. Qiu et al. Issued an anonymous questionnaire with 29 questions and used statistical chi square test to identify the demographic respondents who supported the economic slowdown policy or asked the government to take more measures to combat haze pollution. The results show that people aged between 31 and 50, as well as those living in high pollution areas, support the economic slowdown policy [16]. In order to study and identify the potential impact of socio-economic development on $PM_{2.5}$ change, Qiu et al. Used spatial regression and geographic detector technology to evaluate the correlation direction and intensity between socio-

economic factors and $PM_{2.5}$ concentration using the data of 945 monitoring stations in 190 cities in China in 2014 [17]. Taking 31 provinces in China as research objects, Shi et al. Used the logarithmic mean decomposition index method to decompose the decoupling elasticity coefficient into scale decoupling elasticity, technical decoupling elasticity and structural decoupling elasticity, and analyzed the decoupling relationship between economic growth and environmental pollution from 2001 to 2015 [18]. Zhu et al. Used panel data of 73 key cities in China from 2013 to 2017, and used VECM, impulse response function, variance decomposition and other methods to conduct empirical estimation. The results show that there is a bidirectional causal relationship between $PM_{2.5}$ and economic growth [19]. Based on a large sample of 249 cities in China in 2015, Xie et al. Studied the impact of economic growth on $PM_{2.5}$ pollution using semi parametric spatial autoregressive model [20]. Yang et al. Selected Chengdu Plain Economic Zone (CPEZ), an inland area with severe haze, to identify the spatiotemporal distribution characteristics of $PM_{2.5}$ concentration and its underlying socio-economic factors from 2006 to 2016 by using spatial econometric methods. According to the spatial Durbin model (SDM), socio-economic factors such as population density, energy consumption per unit output, gross domestic product (GDP), per capita GDP and other socio-economic factors have a positive impact on $PM_{2.5}$ concentration [21].

A novel economic impact model is proposed by this paper to analyze the impact of economic downturn on the air quality in Wuhan during the epidemic period, and to explore the effective solutions to improve the urban air pollution. In the past, researchers in related fields often focused on the impact of economic growth on air quality. In the case of almost zero economic variables or negative economic growth, due to the lack of suitable cases, there is no research opportunity. However, the Wuhan case provides researchers with a rare but real special experiment to investigate the impact of economic variables on air quality. In terms of research methods, this paper will theoretically explore how to expand the plane Steiner Weber point into a spatial point, and establish the optimal aggregation method of multi index attribute space and the evaluation method of air quality. By using this method, the comprehensive air quality of Wuhan city from January to April in 2020 and the same period of the previous year in normal years were compared to evaluate the comprehensive air quality when the economic variables were close to zero and the economic variables in normal years were greater than zero, respectively, in order to investigate the impact of economic factors on urban air quality. In addition, this paper also attempts to give the influence degree model of each economic factor on the main pollutants, and estimate the influence degree of simple economic factors on each pollutant, to determine the influence proportion of pure economic factors on air pollutants. In addition to economic factors, indirect estimation of urban air pollution factors, such as geographical location, climate, and other natural factors, as well as the impact of other provinces and regions of imported pollution. It provides valuable reference for the city how to adjust the economic development speed and total amount, promote the transformation of industrial structure in various regions.

## Contributions

The existing research only shows that there is an inevitable relationship between economic development and air pollution, and does not analyze the extent to which various pollutants are affected by economic development. The SOMA proposed by this paper modeling the air quality data of Wuhan in the same period of 2019 and 2020. Then, it is analyzed that the air pollutant most affected by Wuhan's economic development is $NO_2$, which gives guiding suggestions for Wuhan's industrial adjustment and transformation decisions. In addition, the SOMA can be employed for analyzing the air pollution data of any city, according to the development

characteristics of the city, the impact of its economic development on various air pollutants is obtained, and then suggestions on air pollution control are given.

## Methodology

The aggregation value of air multi index data at each time node in a certain period is helpful to evaluate the air quality of the city effectively. Most of the traditional aggregation methods are extended to the ordered weighted averaging (OWA) operator and the induced ordered weighted averaging (IOWA) operator proposed by Yager et al. [22]. However, since a large number of continued products are used in the formulas, the aggregation accuracy of these operators is low when they aggregate the actual air multi index data. Inspired by Steiner Weber point problem, this paper establishes the space optimal aggregation model (SOAM) to aggregate air multi index data.

Steiner-Weber point problem originated from Fermat problem proposed by Fermat in 1643: when a triangle is known, find a point so that the sum of its distances from the three vertices of the triangle is minimal. The solution of this problem is often called Fermat-Torricelli point because Torricelli first solved the problem by using elementary geometric method [23]. Steiner and Weber substantially extended Fermat-Torricelli point, which is called Steiner-Weber point.

### Definition 1

There are $m(m \geq 3)$ given points in a bounded closed box $R^2$ in a two-dimensional plane, whose corresponding positive weights are $\alpha_i \in [0,1](1 \leq i \leq m)$, and $\sum_{i=1}^{m} \alpha_i = 1$. If a point $P^*$ exists, whose weighted Euclidean distances to the other given points meet the following condition:

$$\min_x f(x) = \min \sum_{i=3}^{m} \alpha_i \| x - a_i \| \tag{1}$$

where $a_1, a_2, \cdots, a_m \in R^2$, $\alpha_i$ is the Euclidean distance weight, $\| \cdot \|$ is Euclidean vector norm. Then the optimal aggregation point $P^*$ can be defined as the Fermat-Torricelli point.

A novel aggregation model named Space Optimal Aggregation Model (SOAM) is proposed by this paper by extending the plane Steiner-Weber point model to the space Steiner-Weber point model for solving the multi-index optimal aggregation problem.

### Definition 2

There are $m(m \geq 3)$ weighted points $A_i(x_{i1}, x_{i2}, \cdots, x_{in})(1 \leq i \leq m)$ in a bounded closed box $R^n$ in a $n$-dimensional space, whose corresponding positive weights are $\alpha_i \in [0,1](1 \leq i \leq m)$, and $\sum_{i=1}^{m} \alpha_i = 1$. If a point $B^*(x_1^*, x_2^*, \cdots, x_n^*)$ exists, whose weighted Euclidean distances to the other given points meet the following condition:

$$\min \alpha_i |A_i B^*| = \min \sum_{i=1}^{m} \alpha_i \left( \sum_{j=1}^{n} (x_j^* - x_{ij})^2 \right)^{\frac{1}{2}} \tag{2}$$

Then the optimal aggregation point $B^*$ can be defined as the $n$-dimensional space Fermat-Torricelli point. As shown in Fig 1, take three-dimensional space Fermat-Torricelli point as an example.

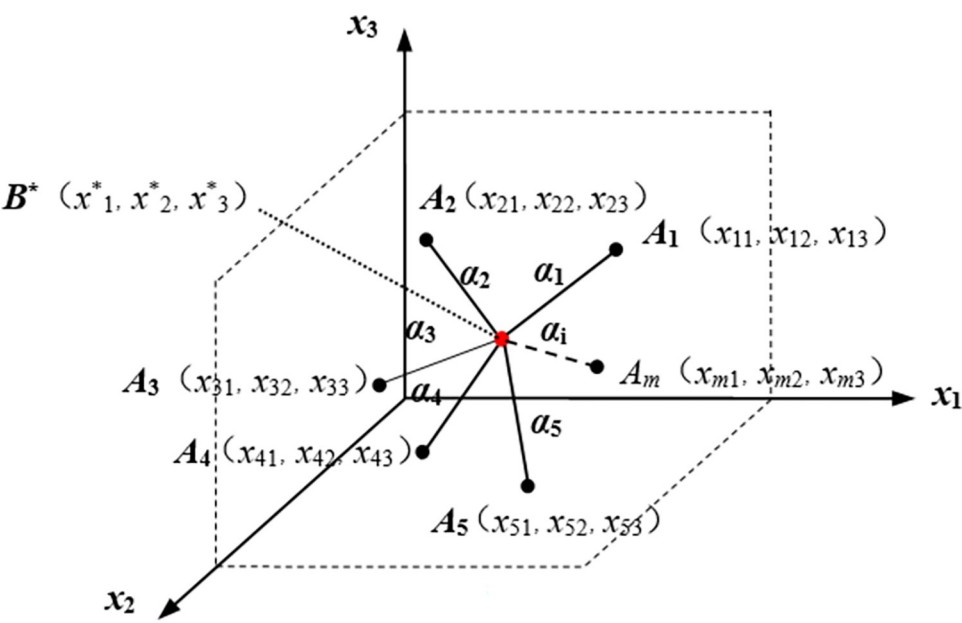

**Fig 1. The three-dimensional space Fermat-Torricelli point.**

When the distance weight is treated as equal, Formula (2) can be rewritten as

$$\min|A_i B^*| = \min \sum_{i=1}^{m} \left( \sum_{j=1}^{n} (x_j^* - x_{ij})^2 \right)^{\frac{1}{2}} \tag{3}$$

In practical applications, Formulas (2) or (3) may be considered as appropriate according to specific circumstances.

The proposed model is used to discuss the optimal aggregation problem of urban air quality in this paper. The air quality of a city in $q$ months are employed as the research object, $m$ days air samples are taken every month to monitor $n$ air quality index data. The air quality data set can be represented by the matrices as follows.

$$A^1 = \begin{pmatrix} x_{11}^1 & x_{12}^1 & \cdots & x_{1n}^1 \\ x_{21}^1 & x_{22}^1 & \cdots & x_{2n}^1 \\ \vdots & \vdots & x_{ij}^1 & \vdots \\ x_{m1}^1 & x_{m2}^1 & \cdots & x_{mn}^1 \end{pmatrix} A^2 = \begin{pmatrix} x_{11}^2 & x_{12}^2 & \cdots & x_{1n}^2 \\ x_{21}^2 & x_{22}^2 & \cdots & x_{2n}^2 \\ \vdots & \vdots & x_{ij}^2 & \vdots \\ x_{m1}^2 & x_{m2}^2 & \cdots & x_{mn}^2 \end{pmatrix} \cdots A^k = \begin{pmatrix} x_{11}^k & x_{12}^k & \cdots & x_{1n}^k \\ x_{21}^k & x_{22}^k & \cdots & x_{2n}^k \\ \vdots & \vdots & x_{ij}^k & \vdots \\ x_{m1}^k & x_{m2}^k & \cdots & x_{mn}^k \end{pmatrix} \tag{4}$$

where $x_{ij}^k$ is the $j$th air quality index data on the $i$th day of the $k$th month. The Matrices are mapped to $q$ $n$-dimensional spaces $E^{n_k}$.

According to the above mapping relationship, the corresponding $q$ $n$-dimensional space coordinate systems can be constructed. The space point set is composed of air quality monitoring index data. Thus, the aggregation model of air quality monitoring index data based on space Steiner-Weber point is established.

Suppose $P_i^k = (x_{i1}^k, x_{i2}^k, \cdots, x_{in}^k)$ as the space point of $i$th day of the $k$th month, then, the space point set of the $k$th month can be expressed as $(P_1^k, P_2^k, \cdots, P_m^k)$, if a point $P^{*k}$ exists,

whose weighted Euclidean distances to the other points in $P_i^k$ meet the condition as follows.

$$d_k = \min\sum_{i=1}^{m} \|P_i^k P^{*k}\| = \min\sum_{i=1}^{m} (\sum_{j-1}^{n} (x_{ij}^k - x_j^{*k})^2)^{\frac{1}{2}} \tag{5}$$

where $\alpha_i$ is the Euclidean distance weight from $p^{*k}$ to $P_i^k$. Then, $p^{*k}$ can be defined as the optimal air quality monitoring index data aggregation point of $P_i^k$. The optimal air quality monitoring index data aggregation matrix $R^*$ can be established by using the optimal air quality monitoring index data aggregation point set $P^* = (P^{*1}, P^{*2}, \cdots, P^{*q})$. $R^*$ can be expressed as follows.

$$R^* = \begin{pmatrix} x_1^{*1} & x_2^{*1} & \cdots & x_n^{*1} \\ x_1^{*2} & x_2^{*2} & \cdots & x_n^{*2} \\ \vdots & \vdots & \vdots & \vdots \\ x_1^{*q} & x_2^{*q} & \cdots & x_n^{*q} \end{pmatrix} \tag{6}$$

When $n = 2$, $m = 3$, the calculating process of $p^{*k}$ is equivalent to solving the plane Fermat-Torricelli point problem, and it can be solved by elementary geometry method. When $n = 2$, $m > 3$, the calculating process of $p^{*k}$ is equivalent to solving the plane Steiner-Weber point problem, and it can be solved by using barycenter method [24, 25]. When $n > 2$, $m > 3$, The aggregation of $p^{*k}$ extends to multidimensional space. At present, there is little research on the algorithm of optimal set node in multidimensional space. In order to solve this problem, the plant growth simulation algorithm (PGSA) is employed by this paper for calculating the optimal air quality monitoring index data aggregation point.

PGSA is an intelligent optimization algorithm using the plant-to-light mechanism as a heuristic criterion, which was proposed by Li et al. in 2005 [26, 27]. This algorithm takes the solution space of the optimization problem as the growth environment of the plant, and uses the optimal solution as the light source. According to the photo-light characteristics of real plants, the deductive mode of the branches and leaves growing rapidly toward the sunlight under different light intensity environments. Fig 2 shows a schematic diagram simulating the photo-growth of plants. The PGSA has attracted the attention of numerous scholars both at home and abroad, who have applied it to respective research fields, the results of which have turned out to be much better than those of other intelligent algorithms [28, 29] Flow chart of the methodology of PGSA can be expressed as follows (see Fig 3).

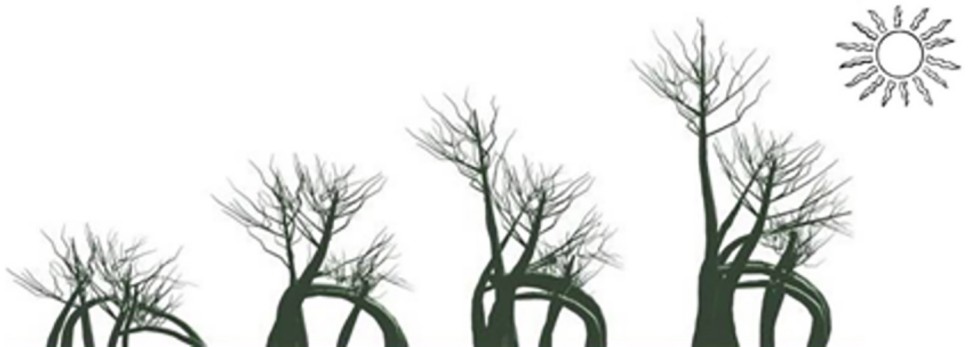

**Fig 2. The schematic diagram of PGSA.**

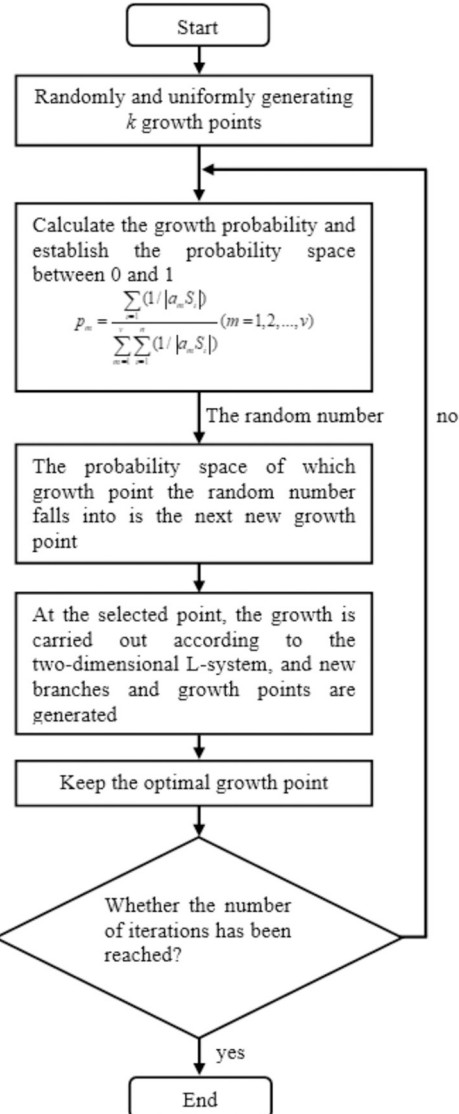

**Fig 3. The flow chart of the methodology of PGSA.**

In order to determine the attribute weight of each air quality index and scheme evaluation, the characteristic root method is employed by this paper for determining the attribute weight of each attribute index [30].

For the air quality test data with $m$ samples and $n$ indexes in each sample, a $m \times n$ order matrix $r$ is formed. Then, the correlation coefficient matrix of $r$ and its characteristic roots $\lambda_j$ are obtained. Thus, the attribute weight of each index $w_j$ can be calculated as follows.

$$w_j = \lambda_j \bigg/ \sum_{j=1}^{n} \lambda_j \tag{7}$$

The linear weighting method was used to evaluate the air quality of each month. The air quality comprehensive evaluation value matrix of $n$ months $\varphi(A)$ can be established as follows.

$$\varphi(A) = W \cdot (R^*)^{\mathrm{T}} \tag{8}$$

where $\varphi(A)$ is the comprehensive evaluation value vector, $A = (A_1, A_2, \cdots, A_q)$ is the evaluation object vector, $W = (w_1, w_2, \cdots, w_n)$ is the attribute weight vector, $R^*$ is the optimal aggregation matrix.

## Case study

Wuhan is the largest city in Central China, and also an important industrial, scientific and educational base and comprehensive transportation hub. In recent years, the momentum of economic development is strong. In 2019, Wuhan city is listed as a new first tier city, with a GDP of 1715.7 billion yuan. However, with the rapid development of economy, the air pollution in Wuhan is serious in recent years, just like many other cities. Due to its special terrain, meteorological conditions, economic and energy consumption growth, Wuhan metropolitan area has become one of the most important air pollution cities in recent years [31]. According to the 2016 World Health Organization's ranking of global air pollution cities, among 210 cities at prefecture level and above in China, Wuhan ranked 12th among the cities with the worst air quality. Its high-density urban spatial form and serious atmospheric particulate matter pollution reflect the typical pollution characteristics of large cities [32]. However, from January 23, 2020 to April 8, 2020, due to the impact of covid-19 virus, in order to prevent the spread of the virus to other areas and the further deterioration of the epidemic situation, Wuhan has implemented unprecedented and most stringent home isolation measures. This sudden major public health emergency made a mega city with a population of 10 million in a relatively high speed of economic development, suddenly forced to press the economic pause key, and the economy was almost in a state of stagnation within more than two months. In order to investigate the impact of economic changes on the local air quality since the outbreak of covid-19 virus in Wuhan, $PM_{2.5}$, $SO_2$ and $NO_2$ were selected as air pollution indicators according to the characteristics of air pollution in Wuhan. Based on the daily monitoring data of $PM_{2.5}$, $SO_2$ and $NO_2$ in Wuhan from January 2020 to April 2020 and the same period of the previous year from http://aqicn.org/city/wuhan/cn/, the model proposed in this paper is used to evaluate and compare them. Through the study of this case, the law of influence on air pollution is revealed when the economic stagnation or negative growth.

The daily monitoring data of $PM_{2.5}$, $SO_2$ and $NO_2$ in Wuhan from January 2020 to April 2020 and the same period of the previous year are aggregated by using Formula (5). The optimal aggregation coordinates $P^{*k}(PM_{2.5}^{*k}, SO_2^{*k}, NO_2^{*k})$ represent the monthly air pollution index data. Thus, the optimal aggregation matrix $R^*$ can be established by employed $P^* = (P^{*1}, P^{*2}, \cdots, P^{*q})$ (see Figs 4–7). Then, Attribute weight vector $W$ and comprehensive evaluation value vector $\varphi$ of each index can be calculated by using Formula (7) and Formula (8) (see Table 1).

The lower the comprehensive index of air pollution is, the better the air quality is. Thus, the order of air quality in different months is as follows.

The first four months of 2020:

$$\varphi(A_4') \succ \varphi(A_3') \succ \varphi(A_2') \succ \varphi(A_1')$$

The first four months of 2020 and the same period of 2019:

$$\varphi(A_1') \succ \varphi(A_1); \varphi(A_2') \succ \varphi(A_2); \varphi(A_3') \succ \varphi(A_3); \varphi(A_4') \succ \varphi(A_4)$$

The air quality from January to April in 2020 is better than that in the same period in 2019. This shows that the air quality of Wuhan is significantly higher than that of the same period of the previous year in the period of economic stagnation and decline caused by the COVID-19 virus epidemic, and the economic factors have a significant impact on the air quality. From

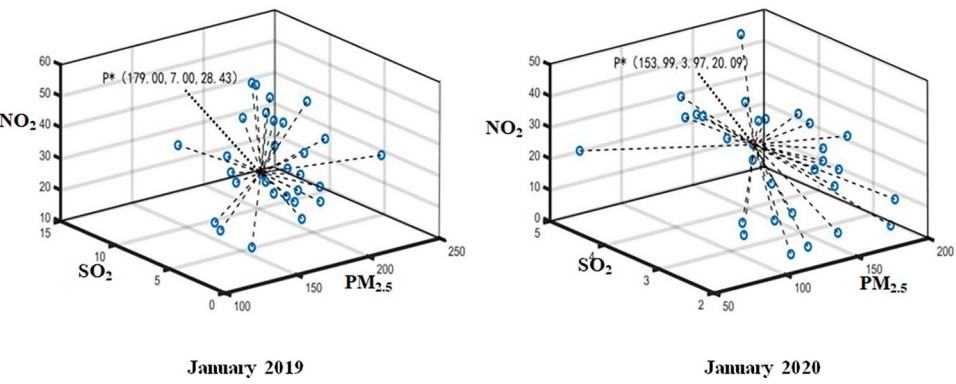

**Fig 4. Aggregation result in January 2019 and 2020.**

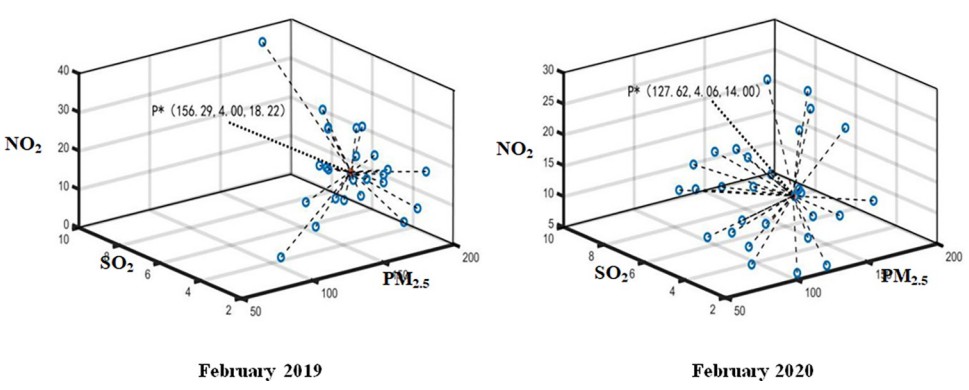

**Fig 5. Aggregation result in February 2019 and 2020.**

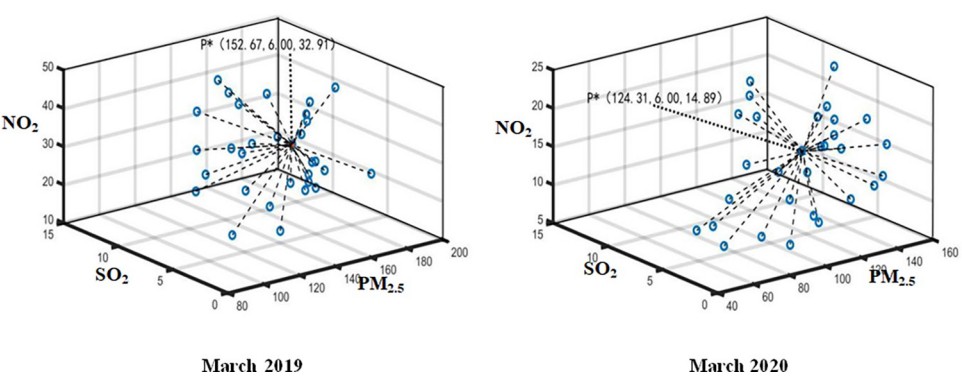

**Fig 6. Aggregation result in March 2019 and 2020.**

January to April in 2020, the comprehensive evaluation value shows a downward trend, that is, the air quality generally presents a good trend. This is related to the city's most stringent home isolation measures in history (see Fig 8).

The data of different pollutants during home isolation in 2020 with the same period of the previous year are compared in this paper in order to further investigate the impact of economic stagnation and decline caused by home isolation measures on $PM_{2.5}$, $SO_2$ and $NO_2$ in Wuhan (see Figs 9–11).

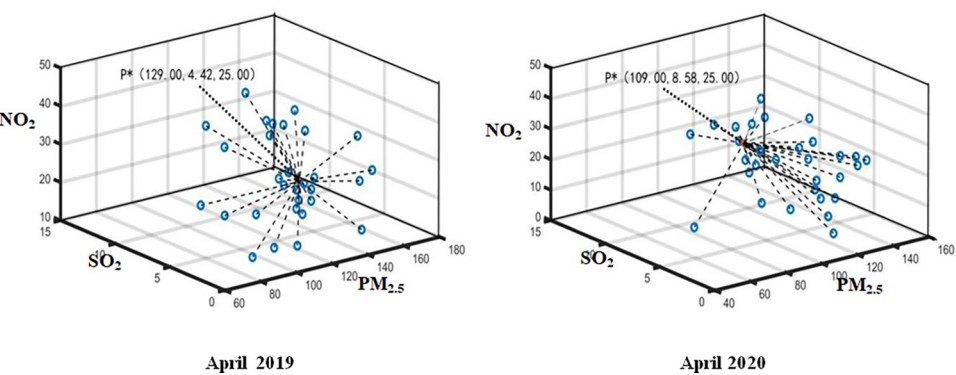

Fig 7. Aggregation result in April 2019 and 2020.

In order to analyze the impact of simple economic factors on various major air pollutants, the calculation formula of economic impact degree is established as follows.

$$E_i = \frac{\overline{PT_i} - \overline{PT_i'}}{\overline{PT_i}} \qquad (9)$$

where $E_i$ is the impact of economy on pollutant $i$ ($i = 1,2,3$; 1 means $PM_{2.5}$, 2 means $SO_2$, 3 means $NO_2$), $\overline{PT_i'}$ is the mean value of the $i$th pollutant during home isolation, and $\overline{PT_i}$ is the mean value of the $i$th pollutant in the same period of last year. The results are as follows.

$$E_1 = \frac{151.59 - 123.51}{151.59} = 19\%; \ E_2 = \frac{5.56 - 4.92}{5.56} = 12\%; \ E_1 = \frac{27.21 - 13.92}{27.21} = 49\%$$

The results show that the influence of economic factors on $PM_{2.5}$ is 19%, that of $SO_2$ is 12%, and that of $NO_2$ is 49%. This result can also be understood as: eliminating economic factors or

Table 1. Calculation results.

| Evaluation object | PM_{2.5} | SO_2 | NO_2 | Attribute weight vector $W$ | The optimal aggregation coordinates | comprehensive evaluation value $\varphi$ |
|---|---|---|---|---|---|---|
| | $\lambda_1$ | $\lambda_2$ | $\lambda_3$ | | $(PM^*_{2.5}, SO^*_2, NO^*_2)$ | |
| $A_1$ (Jan, 2019) | 1.932 | 0.806 | 0.262 | (0.644,0.269,0.087) | (179.00,7.00,28.43) | 119.63 |
| $A_1'$ (Jan, 2020) | 1.829 | 0.818 | 0.353 | (0.610,0.273,0.117) | (153.99,3.97,20.09) | 97.37 |
| $A_2$ (Feb, 2019) | 2.033 | 0.757 | 0.210 | (0.678,0.252,0.070) | (156.29,4.00,18.22) | 108.25 |
| $A_2'$ (Feb, 2020) | 1.847 | 0.708 | 0.445 | (0.616,0.236,0.148) | (127.62,4.06,14.00) | 81.64 |
| $A_3$ (Mar, 2019) | 1.400 | 1.052 | 0.548 | (0.470,0.348,0.182) | (152.67,6.00,32.91) | 79.83 |
| $A_3'$ (Mar, 2020) | 1.849 | 0.731 | 0.420 | (0.613,0.247,0.140) | (124.31,6.00,14.89) | 79.77 |
| $A_4$ (Apr, 2019) | 1.639 | 0.923 | 0.438 | (0.546,0.308,0.146) | (129.00,4.42,25.00) | 75.30 |
| $A_4'$ (Apr, 2020) | 1.572 | 0.977 | 0.451 | (0.524,0.326,0.150) | (109.00,8.58,25.00) | 63.75 |

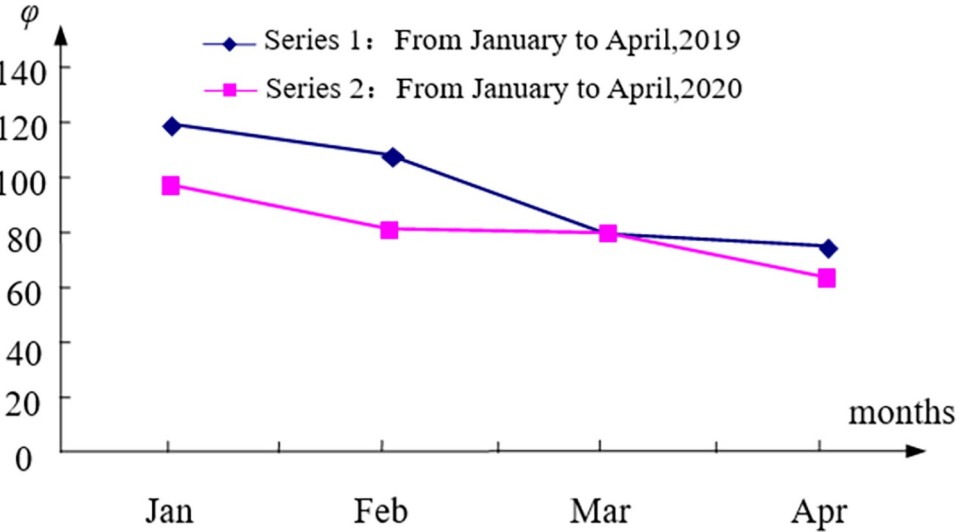

**Fig 8. The change trend of comprehensive evaluation value of air quality.**

increasing economic factors can reduce or increase $PM_{2.5}$, $SO_2$ and $NO_2$ by 19%, 12% and 49%, respectively. The influence degree of pure economic factors on each pollutant is: $NO_2$, $PM_{2.5}$, $SO_2$. In addition to the simple economic factors, other factors affected $PM_{2.5}$, $SO_2$ and $NO_2$ by 81%, 88% and 51% respectively.

The SOAM is suitable for analyzing the relationship between economic development and air pollutants in large and medium-sized cities. The industrial structure of megacities (Shanghai, Beijing, Chongqing, etc.) is complex, and the industrial centers are concentrated (similar enterprises are concentrated in specific regions, while different enterprises are distributed in different regions far away). The air pollution situation in different regions is quite different, the SOAM cannot accurately analyze the relationship between the economic development and air pollutants of the whole city. To solve this problem, megacities can be divided into regions according to the distribution of enterprises, then analyze the air pollution data of different

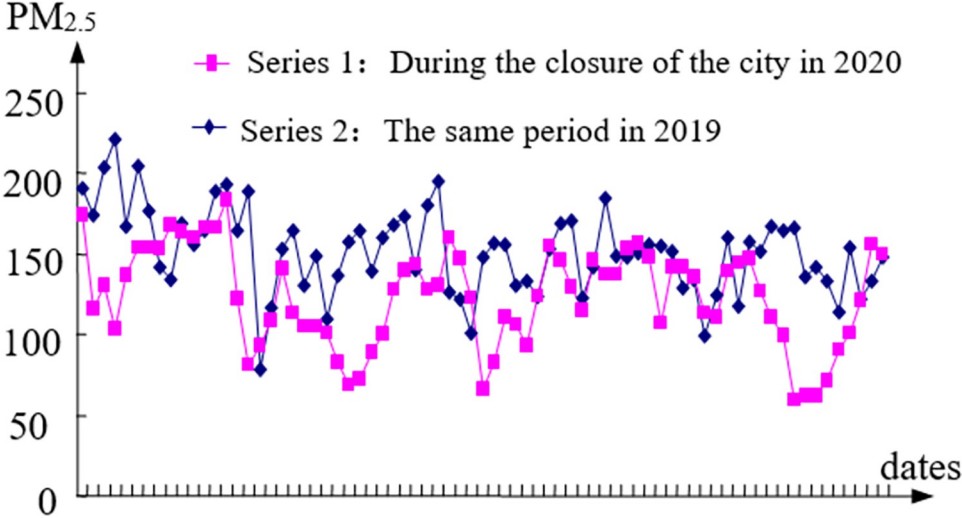

**Fig 9. The comparison of $PM_{2.5}$.**

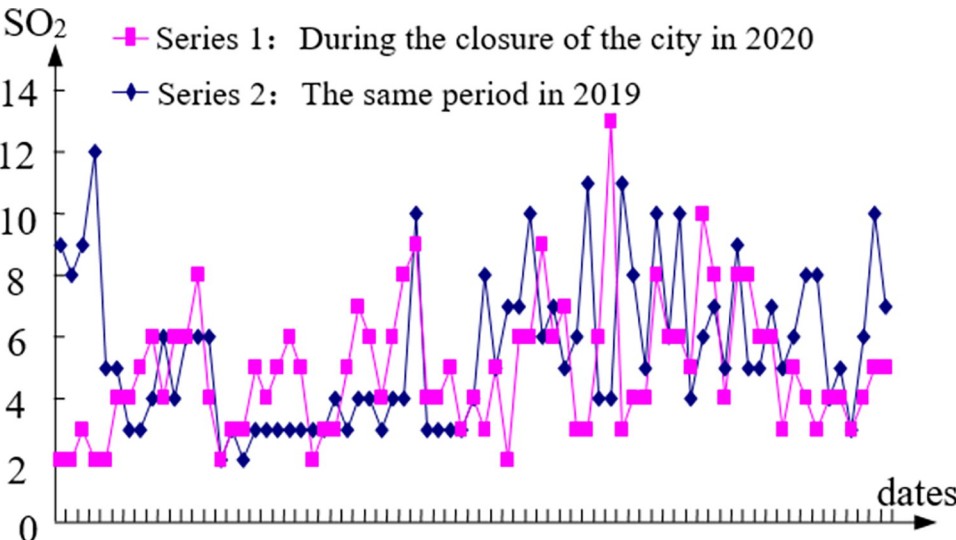

**Fig 10. The comparison of SO$_2$.**

regions. In addition, small cities have single industrial structure and are greatly affected by regional development, so their air pollution is not referential. To solve this problem, multiple small cities in the region can be merged into one urban agglomeration, then adjust the industrial structure by analyzing the relationship between the economic development and air pollution of the urban agglomeration.

## Conclusion

The SOAM aggregation method is proposed in this paper based on the expansion of Steiner-Weber point problem into $n$-dimensional space, and PGSA algorithm is applied to solve the problem that the daily data of main air pollutants with multi-attribute are aggregated into

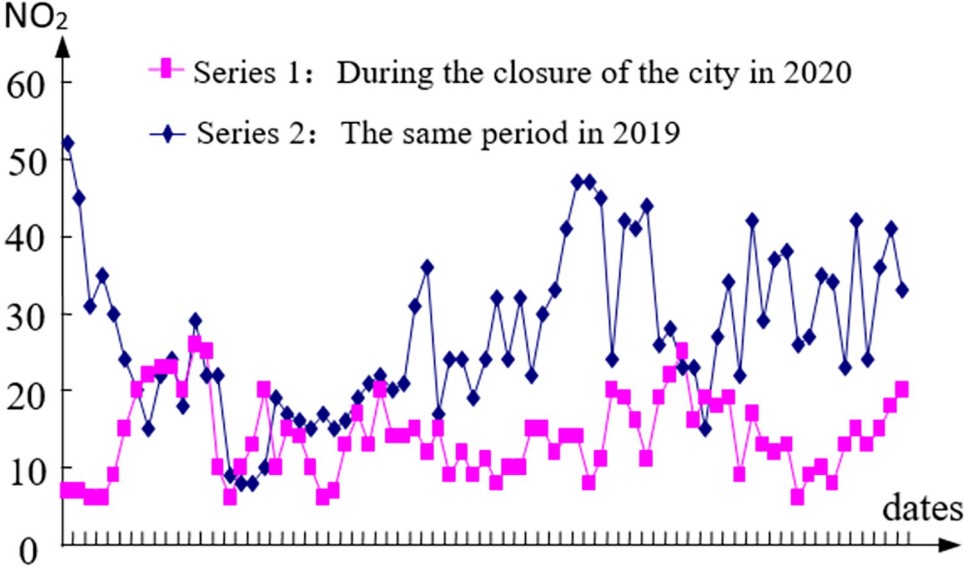

**Fig 11. The comparison of NO$_2$.**

monthly report data. Thus, the comprehensive evaluation value $\varphi$ of air quality in each month can be calculated by linear weighting method, so as to evaluate the air quality of each month.

The air quality of Wuhan city from January to April in 2020 and from January to April in the previous year in each month were studied and evaluated by using SOAM method. The results show that, in the first four months of 2020, after the COVID-19 epidemic situation forced Wuhan to take measures of shutdown, production suspension and home isolation, the economy was in a state of stagnation or decline, and the air quality of Wuhan showed an obvious trend of improvement. Moreover, the overall air quality in the first four months of 2020 is better than that in the same period of 2019 in the normal year of the previous year.

Because of home isolation, the variables of economic factors in the city tend to zero, which provides a very rare real experimental study for the influence of simple economic variables on air quality. In this paper, an economic impact degree model is proposed, which is compared with the same period of 2019 in normal economic activity year to obtain the economic impact degree of major pollutants. The results show that the influence of economic factors on $PM_{2.5}$ is 19%, that of $SO_2$ is 12%, and that of $NO_2$ is 49%. That is to say, eliminating economic factors or increasing economic factors can reduce or increase $PM_{2.5}$, $SO_2$ and $NO_2$ by 19%, 12% and 49% respectively. The influence degree of pure economic factors on each pollutant is: $NO_2$, $PM_{2.5}$, $SO_2$.

According to the data analysis results, we can conclude that Wuhan should urge relevant enterprises to control $NO_2$ emissions through technological innovation and industrial transformation, so as to complete air pollution control. The SOAM proposed by this paper is suis for analyzing the relationship between economic development and air pollution in other large and medium-sized cities in China. According to the degree of correlation between economic development and air pollutants, the core problems faced by local industrial transformation can be accurately obtained.

## Supporting information

**S1 Appendix. PGSA algorithm steps and principles.**
(DOCX)

## Author Contributions

**Data curation:** Junda Qiu.

**Formal analysis:** Junda Qiu, Peng Li.

**Funding acquisition:** Peng Li.

**Investigation:** Junda Qiu.

**Methodology:** Junda Qiu.

**Project administration:** Congzhe You.

**Resources:** Congzhe You.

**Software:** Congzhe You.

**Supervision:** Honghui Fan.

**Validation:** Honghui Fan.

**Visualization:** Honghui Fan.

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
