## [Decision Letter · Decision Letter 0]

1 Nov 2022

PONE-D-22-28609Research of the impact of economic decline on air quality in Wuhan under COVID-19 epidemicPLOS ONE

Dear Dr. Li,

Thank you for submitting your manuscript to PLOS ONE. After careful consideration, we feel that it has merit but does not fully meet PLOS ONE’s publication criteria as it currently stands. Therefore, we invite you to submit a revised version of the manuscript that addresses the points raised during the review process.

We look forward to receiving your revised manuscript.

Kind regards,

Zhihong (Arry) Yao, Ph.D.

Academic Editor

PLOS ONE

Journal Requirements:

"This work was supported in part by the National Natural Science Foundation of China under Grant 61806088, 62002142 and Grant 61902160, in part by the Science and Technology Plan Project of Changzhou under Grant CJ20220055, in part by the Natural Science Foundation of Jiangsu Province under Grant BK20201057."

"Include this sentence at the end of your statement: The funders had no role in study design, data collection and analysis, decision to publish, or preparation of the manuscript."

Reviewers' comments:

Reviewer's Responses to Questions

**Comments to the Author**

1. Is the manuscript technically sound, and do the data support the conclusions?

Reviewer #1: Yes

Reviewer #2: Partly

2. Has the statistical analysis been performed appropriately and rigorously? 

Reviewer #1: Yes

Reviewer #2: Yes

3. Have the authors made all data underlying the findings in their manuscript fully available?

Reviewer #1: Yes

Reviewer #2: Yes

4. Is the manuscript presented in an intelligible fashion and written in standard English?

Reviewer #1: No

Reviewer #2: Yes

5. Review Comments to the Author

Reviewer #1: While your manuscript has apparently undergone some sort of editorial screening and considered to be of sufficient quality to be sent out for review, I'm still finding many problems with the grammar and organization that would lead me to suggest that it still needs editing by someone with some subject area experience.

While the topic is good and the analysis is well-done, the importance of the research needs to be better explained.

I will suggest to the editors that the article be accepted after some major editing that addresses the organization and presentation of the paper.

Reviewer #2: The impact of economic recession on urban air quality under the background of COVID-19 is a good research topic. The author used a novel economic impact model to analyze the impact of economic recession on air quality in Wuhan, and used the Model (SOAM) to evaluate the air quality in Wuhan. Quantitative results are obtained. Overall, the paper is well written, with clear language and logical structure, and it is recommended to be published in“PLOS ONE”with appropriate revisions.

Specific comments：

1."Abstract" further summarizes the main conclusions of the paper and highlights the research focus.

2.The "Introduction" contains a lot of content, and it is suggested to further condense and integrate relevant content to illustrate the innovation and necessity of this research.

3.In "Methodology", it is recommended to add applicability analysis and uncertainty analysis of Model (SOAM).

4.The formula in“L180”is missing number,, please supplement.

5.We all know that recession-induced improvements in air quality are inevitable. Based on the research results of this paper, can the author draw some concrete suggestions on the synergistic development relationship between economy and air quality, so as to make the conclusion of this paper more valuable for reference?

It is suggested to edit the article and adjust the logic of language expression.

6. PLOS authors have the option to publish the peer review history of their article (what does this mean?). If published, this will include your full peer review and any attached files.

Reviewer #1: No

Reviewer #2: No

---

## [Author Response · Author response to Decision Letter 0]

4 Feb 2023

Reviewer #1:

1. Response to comment: (I'm still finding many problems with the grammar and organization that would lead me to suggest that it still needs editing by someone with some subject area experience.)

Response:

Thank you very much for your suggestion. We appreciate your hard work, we are ashamed for our negligence in language inspection. We attach great importance to this opinion. Therefore, we carefully revised the entire manuscript.

2. Response to comment:(While the topic is good and the analysis is well-done, the importance of the research needs to be better explained.)

Response:

 As you suggested that we have revised the abstract, contributions and conclusion to correctly describe the importance of the research.

Revise:

A novel economic impact model is proposed by this paper to analyze the impact of economic downturn on the air quality in Wuhan during the epidemic period, and to explore the effective solutions to improve the urban air pollution. The Space Optimal Aggregation Model (SOAM) is used to evaluate the air quality of Wuhan from January to April in 2019 and 2020. The analysis results show that the air quality of Wuhan from January to April 2020 is better than that of the same period in 2019, and it shows a gradually better trend. This shows that although the measures of household isolation, shutdown and production stoppage adopted during the epidemic period in Wuhan caused economic downturn, it objectively improved the air quality of the city. In addition, the impact of economic factors on PM2.5, SO2 and NO2 is 19%, 12% and 49% respectively calculated by the SOMA. This shows that industrial adjustment and technology upgrading for enterprises that emit a large amount of NO2 can greatly improve the air pollution situation in Wuhan. The SOMA can be extended to any city to analyze the impact of the economy on the composition of air pollutants, and it has extremely important application value at the level of industrial adjustment and transformation policy formulation.

Contributions

The existing research only shows that there is an inevitable relationship between economic development and air pollution, and does not analyze the extent to which various pollutants are affected by economic development. The SOMA proposed by this paper modeling the air quality data of Wuhan in the same period of 2019 and 2020. Then, it is analyzed that the air pollutant most affected by Wuhan's economic development is NO2, which gives guiding suggestions for Wuhan's industrial adjustment and transformation decisions. In addition, the SOMA can be employed for analyzing the air pollution data of any city, according to the development characteristics of the city, the impact of its economic development on various air pollutants is obtained, and then suggestions on air pollution control are given. 

Conclusion

The SOAM aggregation method is proposed in this paper based on the expansion of Steiner-Weber point problem into n-dimensional space, and PGSA algorithm is applied to solve the problem that the daily data of main air pollutants with multi-attribute are aggregated into monthly report data. Thus, the comprehensive evaluation value of air quality in each month can be calculated by linear weighting method, so as to evaluate the air quality of each month.

The air quality of Wuhan city from January to April in 2020 and from January to April in the previous year in each month were studied and evaluated by using SOAM method. The results show that, in the first four months of 2020, after the COVID-19 epidemic situation forced Wuhan to take measures of shutdown, production suspension and home isolation, the economy was in a state of stagnation or decline, and the air quality of Wuhan showed an obvious trend of improvement. Moreover, the overall air quality in the first four months of 2020 is better than that in the same period of 2019 in the normal year of the previous year.

Because of home isolation, the variables of economic factors in the city tend to zero, which provides a very rare real experimental study for the influence of simple economic variables on air quality. In this paper, an economic impact degree model is proposed, which is compared with the same period of 2019 in normal economic activity year to obtain the economic impact degree of major pollutants. The results show that the influence of economic factors on PM2.5 is 19%, that of SO2 is 12%, and that of NO2 is 49%. That is to say, eliminating economic factors or increasing economic factors can reduce or increase PM2.5, SO2 and NO2 by 19%, 12% and 49% respectively. The influence degree of pure economic factors on each pollutant is: NO2, PM2.5, SO2.

According to the data analysis results, we can conclude that Wuhan should urge relevant enterprises to control NO2 emissions through technological innovation and industrial transformation, so as to complete air pollution control. The SOAM proposed by this paper is suitable for analyzing the relationship between economic development and air pollution in other large and medium-sized cities in China. According to the degree of correlation between economic development and air pollutants, the core problems faced by local industrial transformation can be accurately obtained.

Reviewer #2:

1. Response to comment:("Abstract" further summarizes the main conclusions of the paper and highlights the research focus.)

Response:

Thank you very much for your suggestion. We have revised the abstract to further summarizes the main conclusions of the paper and highlights the research focus.

Revise:

A novel economic impact model is proposed by this paper to analyze the impact of economic downturn on the air quality in Wuhan during the epidemic period, and to explore the effective solutions to improve the urban air pollution. The Space Optimal Aggregation Model (SOAM) is used to evaluate the air quality of Wuhan from January to April in 2019 and 2020. The analysis results show that the air quality of Wuhan from January to April 2020 is better than that of the same period in 2019, and it shows a gradually better trend. This shows that although the measures of household isolation, shutdown and production stoppage adopted during the epidemic period in Wuhan caused economic downturn, it objectively improved the air quality of the city. In addition, the impact of economic factors on PM2.5, SO2 and NO2 is 19%, 12% and 49% respectively calculated by the SOMA. This shows that industrial adjustment and technology upgrading for enterprises that emit a large amount of NO2 can greatly improve the air pollution situation in Wuhan. The SOMA can be extended to any city to analyze the impact of the economy on the composition of air pollutants, and it has extremely important application value at the level of industrial adjustment and transformation policy formulation.

2. Response to comment:(The "Introduction" contains a lot of content, and it is suggested to further condense and integrate relevant content to illustrate the innovation and necessity of this research.)

Response:

As you suggested that we have revised the introduction to make it more concise and clear. The research problem statement and contributions are always described in a new section. Your suggestion makes our article more readable.

Revise:

Contributions

The existing research only shows that there is an inevitable relationship between economic development and air pollution, and does not analyze the extent to which various pollutants are affected by economic development. The SOMA proposed by this paper modeling the air quality data of Wuhan in the same period of 2019 and 2020. Then, it is analyzed that the air pollutant most affected by Wuhan's economic development is NO2, which gives guiding suggestions for Wuhan's industrial adjustment and transformation decisions. In addition, the SOMA can be employed for analyzing the air pollution data of any city, according to the development characteristics of the city, the impact of its economic development on various air pollutants is obtained, and then suggestions on air pollution control are given. 

3. Response to comment:(In "Methodology", it is recommended to add applicability analysis and uncertainty analysis of Model (SOAM).)

Response:

As you suggested that we have added applicability analysis and uncertainty analysis of Model in "Methodology".

Revise:

The SOAM is suitable for analyzing the relationship between economic development and air pollutants in large and medium-sized cities. The industrial structure of megacities (Shanghai, Beijing, Chongqing, etc.) is complex, and the industrial centers are concentrated (similar enterprises are concentrated in specific regions, while different enterprises are distributed in different regions far away). The air pollution situation in different regions is quite different, the SOAM cannot accurately analyze the relationship between the economic development and air pollutants of the whole city. To solve this problem, megacities can be divided into regions according to the distribution of enterprises, then analyze the air pollution data of different regions. In addition, small cities have single industrial structure and are greatly affected by regional development, so their air pollution is not referential. To solve this problem, multiple small cities in the region can be merged into one urban agglomeration, then adjust the industrial structure by analyzing the relationship between the economic development and air pollution of the urban agglomeration.

4. Response to comment:(The formula in“L180”is missing number, please supplement.)

Response:

 We are very sorry for our negligence of the number of formula (4). The necessary number is added to all formulas.

5. Response to comment:(We all know that recession-induced improvements in air quality are inevitable. Based on the research results of this paper, can the author draw some concrete suggestions on the synergistic development relationship between economy and air quality, so as to make the conclusion of this paper more valuable for reference?)

Response:

 Your suggestion is very valuable for us. We have added some concrete suggestions on the synergistic development relationship between economy and air quality in "Conclusion".

Revise:

Conclusion

The SOAM aggregation method is proposed in this paper based on the expansion of Steiner-Weber point problem into n-dimensional space, and PGSA algorithm is applied to solve the problem that the daily data of main air pollutants with multi-attribute are aggregated into monthly report data. Thus, the comprehensive evaluation value of air quality in each month can be calculated by linear weighting method, so as to evaluate the air quality of each month.

The air quality of Wuhan city from January to April in 2020 and from January to April in the previous year in each month were studied and evaluated by using SOAM method. The results show that, in the first four months of 2020, after the COVID-19 epidemic situation forced Wuhan to take measures of shutdown, production suspension and home isolation, the economy was in a state of stagnation or decline, and the air quality of Wuhan showed an obvious trend of improvement. Moreover, the overall air quality in the first four months of 2020 is better than that in the same period of 2019 in the normal year of the previous year.

Because of home isolation, the variables of economic factors in the city tend to zero, which provides a very rare real experimental study for the influence of simple economic variables on air quality. In this paper, an economic impact degree model is proposed, which is compared with the same period of 2019 in normal economic activity year to obtain the economic impact degree of major pollutants. The results show that the influence of economic factors on PM2.5 is 19%, that of SO2 is 12%, and that of NO2 is 49%. That is to say, eliminating economic factors or increasing economic factors can reduce or increase PM2.5, SO2 and NO2 by 19%, 12% and 49% respectively. The influence degree of pure economic factors on each pollutant is: NO2, PM2.5, SO2.

According to the data analysis results, we can conclude that Wuhan should urge relevant enterprises to control NO2 emissions through technological innovation and industrial transformation, so as to complete air pollution control. The SOAM proposed by this paper is suitable for analyzing the relationship between economic development and air pollution in other large and medium-sized cities in China. According to the degree of correlation between economic development and air pollutants, the core problems faced by local industrial transformation can be accurately obtained.

---

## [Decision Letter · Decision Letter 1]

13 Feb 2023

PONE-D-22-28609R1Research of the impact of economic decline on air quality in Wuhan under COVID-19 epidemicPLOS ONE

Dear Dr. Li,

Thank you for submitting your manuscript to PLOS ONE. After careful consideration, we feel that it has merit but does not fully meet PLOS ONE’s publication criteria as it currently stands. Therefore, we invite you to submit a revised version of the manuscript that addresses the points raised during the review process.

We look forward to receiving your revised manuscript.

Kind regards,

Zhihong (Arry) Yao, Ph.D.

Academic Editor

PLOS ONE

Journal Requirements:

Reviewers' comments:

Reviewer's Responses to Questions

**Comments to the Author**

1. If the authors have adequately addressed your comments raised in a previous round of review and you feel that this manuscript is now acceptable for publication, you may indicate that here to bypass the “Comments to the Author” section, enter your conflict of interest statement in the “Confidential to Editor” section, and submit your "Accept" recommendation.

Reviewer #1: All comments have been addressed

Reviewer #2: (No Response)

2. Is the manuscript technically sound, and do the data support the conclusions?

Reviewer #1: Yes

Reviewer #2: (No Response)

3. Has the statistical analysis been performed appropriately and rigorously? 

Reviewer #1: I Don't Know

Reviewer #2: (No Response)

4. Have the authors made all data underlying the findings in their manuscript fully available?

Reviewer #1: Yes

Reviewer #2: (No Response)

5. Is the manuscript presented in an intelligible fashion and written in standard English?

Reviewer #1: Yes

Reviewer #2: (No Response)

6. Review Comments to the Author

Reviewer #1: The authors have improved their manuscript. However, I recommend citing more articles on the Covid pandemic, such as：

https://doi.org/10.1007/s41651-020-00064-5

https://doi.org/10.1016/j.atmosres.2020.105362

https://doi.org/10.3390/rs12101576

https://doi.org/10.3390/rs12101613

https://doi.org/10.4209/aaqr.2020.05.0226

Reviewer #2: (No Response)

7. PLOS authors have the option to publish the peer review history of their article (what does this mean?). If published, this will include your full peer review and any attached files.

Reviewer #1: No

Reviewer #2: No

---

## [Author Response · Author response to Decision Letter 1]

13 Feb 2023

Reviewer #1:

1. Response to comment: The authors have improved their manuscript. However, I recommend citing more articles on the Covid pandemic, such as：

https://doi.org/10.1007/s41651-020-00064-5

https://doi.org/10.1016/j.atmosres.2020.105362

https://doi.org/10.3390/rs12101576

https://doi.org/10.3390/rs12101613

https://doi.org/10.4209/aaqr.2020.05.0226

Response:

Thank you very much for your suggestion. We have carefully studied these references and cited them as references for the paper ([6], [7], [10], [32] and [33]).

Journal Requirements:

Response:

Thank you very much for your suggestion. We checked the list of reference list to ensure its validity, and updated some references according to the suggestions of Reviewer #1.

---

## [Editor Report · Decision Letter 2]

20 Feb 2023

Research of the impact of economic decline on air quality in Wuhan under COVID-19 epidemic

PONE-D-22-28609R2

Dear Dr. Li,

We’re pleased to inform you that your manuscript has been judged scientifically suitable for publication and will be formally accepted for publication once it meets all outstanding technical requirements.

Kind regards,

Zhihong (Arry) Yao, Ph.D.

Academic Editor

PLOS ONE
---

## [Editor Report · Acceptance letter]

27 Feb 2023

PONE-D-22-28609R2 

Research of the impact of economic decline on air quality in Wuhan under COVID-19 epidemic 

Dear Dr. Li:

I'm pleased to inform you that your manuscript has been deemed suitable for publication in PLOS ONE. Congratulations! Your manuscript is now with our production department. 

Kind regards, 

on behalf of

Dr. Zhihong (Arry) Yao 

Academic Editor

PLOS ONE